# Factors and impact of physicians' diagnostic errors in malpractice claims in Japan

**Takashi Watari**[1]*, **Yasuharu Tokuda**[2], **Shohei Mitsuhashi**[3], **Kazuya Otuki**[3], **Kaori Kono**[3], **Nobuhiro Nagai**[3], **Kazumichi Onigata**[1], **Hideyuki Kanda**[4]

**1** Postgraduate Clinical Training Center, Shimane University Hospital, Izumo, Shimane, Japan, **2** Okinawa Muribushi Project for Teaching Hospitals, Okinawa, Japan, **3** Shimane University Faculty of Medicine, Izumo, Shimane, Japan, **4** Department of Environmental Medicine & Public Health, Shimane University Faculty of Medicine, Izumo, Shimane, Japan

\* wataritari@gmail.com

## Abstract

### Background

Diagnostic errors are prevalent and associated with increased economic burden; however, little is known about their characteristics at the national level in Japan. This study aimed to investigate clinical outcomes and indemnity payment in cases of diagnostic errors using Japan's largest database of national claims.

### Methods

We analyzed characteristics of diagnostic error cases closed between 1961 and 2017, accessed through the national Japanese malpractice claims database. We compared diagnostic error-related claims (DERC) with non-diagnostic error-related claims (non-DERC) in terms of indemnity, clinical outcomes, and factors underlying physicians' diagnostic errors.

### Results

All 1,802 malpractice claims were included in the analysis. The median patient age was 33 years (interquartile range = 10–54), and 54.2% were men. Deaths were the most common outcome of claims (939/1747; 53.8%). In total, 709 (39.3%, 95% CI: 37.0%–41.6%) DERC cases were observed. The adjusted total billing amount, acceptance rate, adjusted median claims payments, and proportion of deaths were significantly higher in DERC than non-DERC cases. Departments of internal medicine and surgery were 1.42 and 1.55 times more likely, respectively, to have DERC cases than others. Claims involving the emergency room (adjusted odds ratio [OR] = 5.88) and outpatient office (adjusted OR = 2.87) were more likely to be DERC than other cases. The initial diagnoses most likely to lead to diagnostic error were upper respiratory tract infection, non-bleeding digestive tract disease, and "no abnormality."

### Conclusions

Cases of diagnostic errors produced severe patient outcomes and were associated with high indemnity. These cases were frequently noted in general exam and emergency rooms

**Data Availability Statement:** Data cannot be shared publicly based on the contract with Westlaw Japan Inc. However, the data underlying the results presented in the study are completely available

from Westlaw Japan (e-mail address: support@westlawjapan.com, URL of the data base; https://go.westlawjapan.com/wljp/app/signon/display), with an annual subscription. The authors of this study had no special access privileges to the data others would not have.

**Funding:** T.W., Y.T., and H.K. are supported by grants from the National Academic Research Grant Funds (JSPS KAKENHI: 17K15745). The sponsor of the study had no role in the study design, data collection, analysis, or preparation of the manuscript.

**Competing interests:** The authors have declared that no competing interests exist.

as well as internal medicine and surgery departments and were initially considered to be common, mild diseases.

## Introduction

According to the landmark report titled Improving Diagnosis in Health Care, [1] cases of diagnostic errors are common, but it is difficult to measure them; thus, they are frequently overlooked. Several studies have revealed a substantial incidence and economic burden associated with diagnostic error. [1–5] One study estimated the outpatient diagnostic error rate in the US as 5.08%, which translates to approximately 12 million adults per year. [5] Another study estimated that 40,000 to 80,000 patient deaths that occur each year in the US are due to diagnostic errors. [6]

Diagnostic errors can be studied using several means, including data from: 1) malpractice claims, 2) autopsies, 3) questionnaire surveys, 4) case reviews, 5) hospital incident reports, 6) patient surveys, and 7) secondary reviews. [7] Diagnostic errors are a common reason for malpractice claims, [8–15] and claims data could provide vital information from patients' viewpoints. [2,3] For instance, Tokuda and colleagues summarized the findings from 274 malpractice claims filed at two local district courts in Tokyo and Osaka, and found that cognitive errors were the most common errors associated with these medical claims. [12]

However, little is known regarding diagnostic error-related malpractice claims at the national level in Japan. Thus, the objective of the present study was to better characterize the negative impact of diagnostic errors reported in malpractice claims, including the magnitude of indemnity payment and severity of patient outcomes. An additional objective was to compare these effects and the underlying factors of physicians' diagnostic errors between diagnostic error-related claims (DERC) and non-diagnostic error-related claims (non-DERC). Certain background factors, such as clinical specialty or work settings, would be more likely related to DERC cases. Finally, we explored the most frequent initial diagnoses in cases of diagnostic error.

## Methods

### Study design

We conducted a retrospective review of claims data related to medical malpractice cases closed between 1961 and 2017 from the largest database in Japan (Westlaw Japan K.K.), [13] a public-use data file that includes 223,218 Japanese lawsuit cases. This database was used to identify the reported claims, outcomes, and payments for closed claims. While the malpractice claims were anonymous, we were able to extract detailed medical information for each case.

### Study protocol

We used a permuted combination of keywords for claims: medical claims, medical malpractice, medical litigations, diagnostic errors, wrong diagnosis, misdiagnosis, and delayed diagnosis. All claims cases were merged into a single tabular list (3,430 cases). Before extracting the data, the primary investigator and a senior medical student who was also a qualified lawyer set the exclusion criteria: duplications of cases, intentional crimes, robbery, money troubles, and veterinary claims. We excluded 751 cases that were duplicates, 707 cases based on the other exclusion criteria, 34 cases that constituted an "unfair suit," and 136 cases with a non-physician

defendant (nurse = 51 cases, paramedic = 47 cases, "other" = 36 cases). This left us with 1,802 cases to analyze (Fig 1).

Reviewers for the present study constituted five members: one senior medical student who was also a qualified lawyer, one qualified pharmacist, two senior medical students, and a primary investigator. Additionally, the primary investigator trained the four co-investigators. Three Japanese physicians, who were certified by the Japanese Medical Specialty Board and specialized in internal medicine or pediatrics, and one public health professional guided the research team. Finally, all claims reviews were confirmed by the primary investigator.

## Variables and definitions

We included medical providers' characteristics (doctor specialties, clinical settings, day or night shift, and scale of the medical facilities) and patient background (age, sex, dispute point in the cases, treatment styles, initial diagnosis, and final diagnosis). Doctor specialty classification was based on the Japanese Medical Specialty Board (2019). [14] All of the targeted cases were labeled as DERC or non-DERC by the three co-investigators and confirmed by the primary investigator. The most recent definition for a diagnostic error is "the failure to (a) establish an accurate and timely explanation of a patient's health problem(s) or (b) communicate this explanation to the patient." [1] However, to minimize bias during the review, we selected the widely used definitions of a diagnostic error: "delay in diagnosis," "misdiagnosis," and "wrong diagnosis." [15] Judgments were deemed final if made by the Supreme Court, high courts, or local district courts.

## Outcome measures

Our primary outcome variables were deaths, sequelae, full recovery, claims with final judgments, and indemnity amount for the malpractice claims. All payment values were adjusted to the 2017 equivalent using the Japanese Consumer Price Index (available at https://www.stat.go.jp/data/cpi/, Japanese Ministry of Internal Affairs and Communications). Each payment amount was converted from Japanese yen to US dollars ($1 = ¥110; June 1, 2019).

## Statistical analyses

We used standard descriptive statistics to calculate the number, percentage, mean, and median payment amounts for each malpractice claim. A chi-squared test or Fisher's exact test was used to compare nominal variables. For continuous variables, t-tests or Wilcoxon rank sum tests were employed where appropriate. All analyses were performed using Stata statistical software, version. 14.0 (Stata Corp. 2015, Stata 14 Base Reference Manual). All tests were two-sided with $p < 0.05$ considered statistically significant.

## Results

All 1,802 malpractice claims extracted from the database between January 1, 1961 and June 29, 2017 were included in the analysis. In the extracted data, malpractice claim frequency was measured using 10-year periods, and the number of claims in each period was determined: before 1970 (n = 198; 11.0% of total malpractice claims), during the 1970s (n = 393; 21.8%), during the 1980s (n = 366; 20.3%), during the 1990s (n = 623; 34.6%), during the 2000s (n = 182; 10.1%), and during the 2010s (n = 30; 1.7%). Although we collected all available malpractice claim cases from the database, most data represented cases that occurred after the 1970s (n = 1,594; 88%). The DERC percentage for each 10-year period was significantly different only for the period before 1970 and during the 2000s (before 1970: 26.79%, $p < 0.001$;

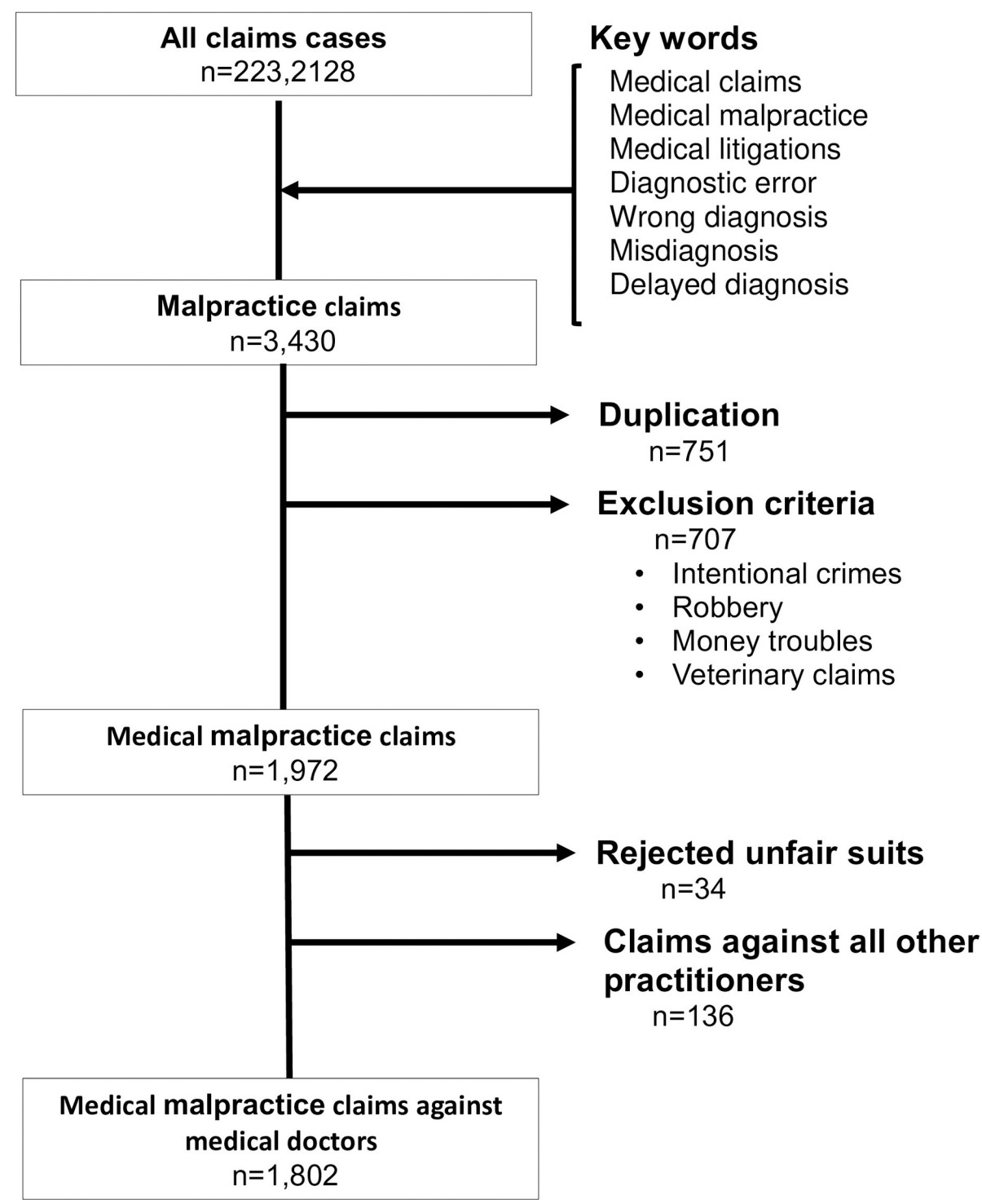

**Fig 1. Flow diagram of malpractice claims for each search strategy.**

2000s: 52.0%, p < 0.001). We also performed a multivariate logistic analysis to compare the DERC and non-DERC groups over each 10-year period and found no significant difference in the proportion of DERC for any period. The basic demographic data for all claims are shown in Table 1. The median patient age was 33 years (interquartile range [IQR] = 10–54), and 54.2% were men. The median adjusted total billing amount was $382,727 (n = 1,802, IQR = $128,182–$909,091), and the adjusted median for the final judgment amount was $183,636 (n = 941, IQR = $41.462–$440,909), with 52.6% of the claims (n = 941) having a final judgment resulting in payment. The median claim duration was 7 years (M = 7.79 years, IQR = 5–10 years, maximum = 28 years). Death was the most common claims outcome (n = 939/1747; 53.8%), followed by sequelae (39.7%) and full recovery (6.5%). A total of 709 claims (39.3%, 95% confidence interval [CI]: 37.0%–41.6%) were DERC cases. In addition, we specifically analyzed the 941 claims where final judgment resulted in payment. Among these, death was the most common claim outcome (n = 473/941; 50.27%), followed by sequelae (41.8%) and full recovery (6.3%). The median patient age was 32 years (interquartile range [IQR] = 10.5–53), and 54.2% were men. The median claim duration was 7 years (M = 7.64 years, IQR = 5–9 years, maximum = 25 years). Of these, 447 claims (47.5%, 95% confidence interval [CI]: 44.3%–50.7%) were DERC cases.

Table 2 provides information on the initial diagnosis for the cases that were categorized into the DERC and non-DERC groups. The two most common initial diagnoses of DERC-involved patients were malignant neoplasms (n = 65; 9.2%) and traumatic injury (n = 64; 8.7%). Further, the five most common malignant diseases were gastric cancer (n = 24; 16%), colorectal cancer (n = 22; 14.7%), breast cancer (n = 16; 10.7%), liver cancer (n = 14; 9.3%), and lung cancer (n = 10; 6.7%). However, these diseases did not occur at a significantly higher frequency than they did in the non-DERC cases.

Respiratory tract infection, no abnormalities, and non-bleeding digestive tract disease were the most common initial diagnoses for DERC cases, in which they were significantly more frequent than in the non-DERC cases (7.1% vs 2.5%, 5.6% vs 0.7%. and 5.2% vs 1.4%, respectively; all p < 0.001). Importantly, upper respiratory infections, such as the common cold, acute bronchitis, and pharyngitis, were the most common diagnostic errors when the initial

**Table 1. Background of malpractice claims in Japan (n = 1,802).**

| Patient sex (male %) | | 54.2% |
|---|---|---|
| Patient age (IQR) | | 33 (IQR 10–54) |
| Adjusted total billing amount ($) | | 382,727 (IQR 128,182–909,091) |
| Claims with final judgment resulting in payment | | 941 (52.6%) |
| Adjusted median accepted settlement amount ($) | | 183,636 (IQR 41,162–440,909) |
| Duration of claim | | 7 years (IQR 5–10) |
| Outcome | | |
| | Deaths | 939 (53.8%) |
| | Sequelae | 694 (39.7%) |
| | Full recovery | 114 (6.5%) |
| | Other | 55 (3.0%) |

Patient demographics and characteristics of claims. The total billing amount and median indemnity were adjusted to the 2017 equivalent using the Japanese Consumer Price Index (showed in USD). IQR: Interquartile range.

**Table 2. The initial diagnosis of DERC and Non-DERC.**

| Initial Diagnosis | DERC | | Non-DERC | | P-value |
|---|---|---|---|---|---|
| | (n = 709) | | (n = 1,093) | | |
| Malignant neoplasm | 65 | 9.2% | 85 | 7.8% | 0.291 |
| Traumatic injury | 62 | 8.7% | 94 | 8.6% | 0.014 |
| Respiratory tract infection | 50 | 7.1% | 27 | 2.5% | < 0.001 |
| No abnormalities | 40 | 5.6% | 8 | 0.7% | < 0.001 |
| Non-bleeding digestive tract disease | 37 | 5.2% | 15 | 1.4% | < 0.001 |
| Ischemic heart disease | 32 | 4.5% | 40 | 3.7% | 0.366 |
| Neonatal complications | 29 | 4.1% | 87 | 8.0% | 0.001 |
| Systematic infectious disease | 27 | 3.8% | 36 | 3.3% | 0.561 |
| Hepatobiliary and pancreatic disease | 24 | 3.4% | 22 | 2.0% | 0.071 |
| Endocrine and metabolic disorders | 23 | 3.2% | 13 | 1.2% | 0.002 |
| Airway and respiratory disorders | 18 | 2.5% | 16 | 1.5% | 0.101 |
| Uterine appendage | 18 | 2.5% | 29 | 2.7% | 0.882 |
| Cerebrospinal disease (except for infectious diseases and strokes) | 17 | 2.4% | 30 | 2.7% | 0.652 |
| Endocrine metabolic disease | 16 | 2.3% | 18 | 1.6% | 0.353 |
| Stroke | 14 | 2.0% | 24 | 2.2% | 0.750 |
| Procedure and post-operative complications | 14 | 2.0% | 65 | 5.9% | < 0.001 |
| Renal and urinary diseases | 13 | 1.8% | 16 | 1.5% | 0.542 |
| Mental disorder | 12 | 1.7% | 45 | 4.1% | 0.004 |
| Appendicitis | 12 | 1.7% | 19 | 1.7% | 0.947 |
| Central nervous system infection | 12 | 1.7% | 5 | 0.5% | 0.008 |
| Dental problems | 2 | 0.3% | 7 | 0.6% | 0.297 |
| Other | 91 | 12.8% | 155 | 14.2% | 0.416 |

The top 20 initial diagnoses involved in malpractice claims in the total number of diagnostic error-related claims (DERC) as compared to non-DERC.

diagnosis was a mild respiratory infection at the time of the first consultation (n = 48/77; 62.3%). Additionally, an initial diagnosis of gastroenteritis (n = 23/52; 44.2%) and intestinal obstruction (n = 21/52; 40.4%) were the two most frequent claims among patients reporting a non-bleeding digestive tract disease.

Table 3 ranks the top 15 clinical departments cited across malpractice claims. The most frequently cited department was internal medicine (n = 216; 30.5%), followed by surgery (n = 150; 21.2%), obstetrics and gynecology (n = 113; 16.1%), pediatrics (n = 54; 7.6%), and orthopedics (n = 45; 6.3%). DERC cases were more likely to occur among physicians working in internal medicine (odds ratio [OR] = 1.86, 95% CI: 1.50–2.32), surgery (OR = 1.28, 95% CI: 1.01–1.63), and emergency medicine (OR = 3.89, 95% CI: 1.29–11.78).

Malpractice payment characteristics and outcomes by diagnostic error are presented in Table 4. Mortality rates were higher for DERC than non-DERC cases in smaller hospitals (23.3% vs. 17.3%; p = 0.002), in general outpatient and emergency rooms (42.9% vs. 15.8%; p < 0.001), and during night shifts when compared to daytime shifts (17.5% vs. 12.0%; p < 0.001). The adjusted total billing amount for DERC cases tended to be higher than that for non-DERC cases. Additionally, the percentages of final judgments for DERC and non-DERC claims were 63.6% and 46.0% (p < 0.001), and the median claims payments for the two types were $231,181 (IQR = $50,150–$484,546) and $136,363 (IQR = $30,554–$370,000), respectively. The mean DERC duration was approximately 6 months shorter than the non-DERC duration. Furthermore, patient outcomes were not trivial. The most common adverse event was death (939 claims; 52.1%; 95% CI, 49.8 to 54.4%). Importantly, the percentage of deaths

**Table 3. Comparison of departments and specialties among malpractice claims.**

| | | DERC | | Non-DERC | | P-value |
|---|---|---|---|---|---|---|
| **Clinical departments of malpractice claims** | | n = 709 | | n = 1,093 | | |
| | Internal Medicine | 216 | 30.5% | 208 | 19.0% | < 0.001 |
| | Obstetrics and Gynecology | 114 | 16.1% | 221 | 20.2% | 0.027 |
| | Surgery | 150 | 21.2% | 189 | 17.3% | 0.04 |
| | Orthopedics | 45 | 6.3% | 117 | 10.7% | 0.002 |
| | Pediatrics | 54 | 7.6% | 69 | 6.3% | 0.284 |
| | Neurosurgery | 32 | 4.5% | 61 | 5.6% | 0.317 |
| | Otolaryngology | 30 | 4.2% | 34 | 3.1% | 0.209 |
| | Ophthalmology | 15 | 2.1% | 44 | 4.0% | 0.026 |
| | Psychiatry | 8 | 1.1% | 39 | 3.6% | 0.002 |
| | Plastic Surgery | 2 | 0.3% | 36 | 3.3% | < 0.001 |
| | Urology | 12 | 1.7% | 13 | 1.2% | 0.372 |
| | Dermatology | 5 | 0.7% | 12 | 1.1% | 0.464 |
| | Emergency Medicine | 10 | 1.4% | 4 | 0.4% | 0.024 |
| | Anesthesiology | 4 | 0.6% | 10 | 0.9% | 0.585 |
| | Radiology | 4 | 0.6% | 4 | 0.4% | 0.719 |
| | Rehabilitation | 2 | 0.3% | 2 | 0.2% | 0.552 |
| | Family Practice | 1 | 0.1% | 1 | 0.1% | 1 |
| | Pathology | 1 | 0.1% | 0 | 0.0% | 0.393 |
| | Other | 6 | 0.8% | 29 | 2.7% | 0.006 |

DERC: Diagnostic error-related claims

among DERC cases was significantly higher than that among non-DERC cases (62.3% vs. 45.5%, p < 0.001).

The results of a multiple logistic regression analysis with the ORs of the characteristics predicting malpractice claims are presented in Table 5. The adjusted model, controlling for type of medical care and initial diagnosis provided, is reported. In this adjusted model, internal medicine departments were 1.42 times more likely to have a DERC than other departments (95% CI: 1.10–1.83, p < 0.007). Surgical departments were 1.55 times more likely to encounter DERC cases than any other department (95% CI: 1.18–2.03, p = 0.001). However, emergency departments were not more likely to experience DERC cases relative to other departments (adjusted OR = 2.84, 95% CI: 0.79–10.2, p = 0.109). Small hospitals were 1.29 times more likely to have DERC cases than any other setting, including clinics and medium-to-large hospitals (95% CI: 1.00–1.67, p = 0.048). Importantly, emergency rooms (adjusted OR = 5.88) and outpatient offices (adjusted OR = 2.87) were much more likely to encounter DERC cases than any other setting (p < 0.001). However, night shifts were not significantly more likely to receive DERC cases relative to daytime shifts (adjusted OR = 1.26, 95% CI: 0.92–1.72; p = 0.146). Notably, if a respiratory tract infection was the initial diagnosis, the likelihood that the case was a DERC case was 2.39 times higher (95% CI: 1.44–4.00; p < 0.001). The likelihood of DERC categorization was a 3.24 times higher for non-bleeding digestive tract disease, with gastroenteritis being the most common initial diagnosis (95% CI: 1.71–6.14, p < 0.001), and 7.07 times higher for a "no abnormality" initial diagnosis (95% CI 3.2–15.61, p < 0.001).

**Table 4. Comparison of the Facility Size, Place, Time of Occurrence, and Results among Malpractice Claims.**

| | | DERC | Non-DERC | P-value |
|---|---|---|---|---|
| | | n = 709 | n = 1,093 | |
| **Facility size** | | | | |
| | **Clinic** | 197 (27.8%) | 300 (27.4%) | 0.875 |
| | **Small hospital** | 165 (23.3%) | 189 (17.3%) | 0.002 |
| | **Medium hospital** | 244 (34.4%) | 364 (33.3%) | 0.626 |
| | **Large hospital** | 93 (13.1%) | 228 (20.9%) | < 0.001 |
| **Place** | | | | |
| | **Outpatient office** | 218 (30.7%) | 150 (13.7%) | < 0.001 |
| | **Ward** | 184 (26.0%) | 324 (29.6%) | 0.089 |
| | **Emergency room** | 86 (12.1%) | 23 (2.1%) | < 0.001 |
| | **Procedure and operation room** | 207 (29.2%) | 575 (52.6%) | < 0.001 |
| **Time** | | | | |
| | **Daytime** | 584 (82.4%) | 960 (87.8%) | 0.011 |
| | **Nighttime** | 124 (17.5%) | 131 (12.0%) | 0.001 |
| **Cases with final court judgment** | | 451 (63.6%) | 503 (46%) | < 0.001 |
| **Adjusted total billing amount** | | 440,000 | 350,909 | < 0.001 |
| | | (IQR 185,454–775.455) | (IQR 107,272–660,000) | |
| **Adjusted total accepted amount** | | 231,181 | 136,363 | < 0.001 |
| | | (IQR 50,150–484,546) | (IQR 305,54–370,000) | |
| **Duration of claim case** | | 7.48 (± 3.68) | 7.99 (± 4.06) | 0.005 |
| **Outcome** | | | | |
| | **Deaths** | 442 (62.3%) | 497 (45.5%) | < 0.001 |
| | **Sequelae** | 219 (30.9%) | < 0.001 | < 0.001 |
| | **Full recovery** | 31 (4.4%) | 83 (7.6%) | 0.006 |

DERC: Diagnostic error-related claims.

## Discussion

Our results, based on 1,802 malpractice claims over a 56-year period in Japan, showed that about 40% of the claims involved allegations of a diagnostic error, that the most frequent outcome was death, and that the magnitude of indemnity payment was variable but its median value was expensive. The initial diagnoses most commonly associated with allegations of diagnostic errors in malpractice claims were upper respiratory tract infection (mainly the common cold), non-bleeding digestive tract disease (mainly gastroenteritis), and "no abnormality." Thus, it is important to improve diagnostic skills to differentiate between life-threatening conditions and an innocuous upper respiratory tract infection [16] or common gastrointestinal disease. [17] Although several studies have examined final diagnoses involving malpractice claims within each clinical specialty, [18–26] few have examined initial inaccurate diagnoses that were later identified as incorrect diagnoses.

Our results from comparisons between DERC and non-DERC cases were similar to those from a study by Gupta et al. [2] indicating that DERC cases are more likely to be associated with death and greater compensation costs, although in other claims studies that focused on errors in limited settings such as emergency department, [25] pediatrics, [20,26] and inpatient-outpatient settings [27,28] in the US, death was reported less frequently (36%, 28.2%, and 30.4%, respectively). In a 25-year summary of DERC malpractice cases in the US, death was

**Table 5. Results of multiple logistic regression analysis of DERC.**

| | Unadjusted OR | P-value | Adjusted OR | P-value |
|---|---|---|---|---|
| | (95% CI) | | (95% CI) | |
| Department of Internal Medicine | 1.86 (1.50–2.32) | < 0.001 | 1.42 (1.10–1.83) | 0.007 |
| Department of Surgery | 1.28 (1.01–1.63) | 0.041 | 1.55 (1.18–2.03) | 0.001 |
| Department of Emergency Medicine | 3.89 (1.22–12.47) | 0.022 | 2.84 (0.79–10.2) | 0.109 |
| Small hospital (beds<100) | 1.45 (1.15–1.83) | 0.002 | 1.29 (1.00–1.67) | 0.048 |
| General exam room | 2.79 (2.21–3.53) | < 0.001 | 2.87 (2.22–3.71) | < 0.001 |
| Emergency room | 6.42 (4.01–10.28) | < 0.001 | 5.88 (3.51–9.83) | < 0.001 |
| Nighttime | 1.56 (1.19–2.03) | 0.001 | 1.26 (0.92–1.73) | 0.146 |
| Initial diagnosis | | | | |
| Respiratory tract infection | 3.00 (1.86–4.83) | < 0.001 | 2.39 (1.44–4.0) | 0.001 |
| Non-bleeding digestive tract disease | 3.96 (2.16–7.27) | < 0.001 | 3.24 (1.71–6.14) | < 0.001 |
| No abnormality | 8.11 (3.77–17.43) | < 0.001 | 7.07 (3.2–15.61) | < 0.001 |

The odds ratios (OR) and 95% confidence intervals (CI) are reported. Department, small hospital size, general exam room, emergency room, night shift, and each selected initial diagnosis (respiratory tract infection, non-bleeding digestive tract disease, or no abnormality) were incorporated in the multiple logistic regression analysis.

DERC: Diagnostic error-related claims

the most common outcome. Further, another Japanese study examining closed malpractice claims observed death as a frequent outcome (45%). [12]

Regarding the specialty of the sued physicians, one study demonstrated that those of certain specialties, such as internal medicine, general surgery, and obstetrics and gynecology, have a higher risk of involvement in malpractice claims. [29] Our results were similar and showed that physicians in emergency or family medicine are less likely to be involved in diagnostic error claims. There are several possible causes for this. First, historically, there are few emergency physicians in Japan who are certified by the Japanese Medical Specialty Board. In fact, general surgeons or internists often work as tentative emergency physicians during nighttime shifts, weekends, or holidays. Second, the system of training and supporting generalists, such as general practitioners or family physicians, has only recently begun in Japan, as it started in 2018. [14] Third, in Japan, most sub-specialists in internal medicine, such as gastroenterologists or cardiologists, commonly change their specialty to become general internal medicine physicians during the middle stage of their careers without receiving any additional training as generalists. Worldwide, Japan has the largest total number of hospitals and beds per national population. Overall, it has more than 4,000 emergency rooms; however, the number of emergency physicians who have been certified by the board of emergency care in Japan is quite low (approximately 4,500). [30] Thus, an imbalance in the number of hospitals and emergency physicians has occurred. Compared to other high-income countries, non-emergency physicians working as surgeons or internists at small- to medium-sized hospitals in Japan are also required to work as emergency physicians, regardless of their specialty. Thus, regarding high-risk specialties, a simple comparison to studies on malpractice claims in other countries is not warranted since Japan has a lower proportion of emergency physicians and family physicians, which are the specialties with the highest risk of involvement in diagnostic errors in other countries. [4,5,27,28] The emergency room in Japan is a unique but high-risk setting where physicians are required to diagnose serious injuries or ailments even though many of them are not trained in emergency medicine. [31] Our results also suggest that DERC risk in outpatient departments is high. In addition, Japanese doctors working in primary care settings, including

surgeons and internists, are forced to make clinical decisions with a high patient load in a limited amount of time. A study on malpractice claims in the Netherlands revealed that diagnostic errors occurred more often during the afternoon and evening (58%). [32] Therefore, further training in general and emergency medicine, along with improvement of work conditions, are required to minimize diagnostic errors in these settings in Japan.

However, several limitations of this study should be noted. First, although we used the largest claims database in Japan (similar to previous studies [12,29,32,33]), the data were not nationally representative of all malpractice claims. Second, there are no existing data on the frequency with which adverse events lead to malpractice claims in Japan. According to the Japanese Supreme Court report, the total number of adjudicated medical lawsuits in Japan, including those heard in brief and district courts, was not large [13]: there were 305 malpractice claims that received their final judgments in 2000, 324 in 2010, and 269 in 2016. Third, the data from malpractice claims are not direct medical records and are thus not ideal sources for investigating error frequency and causal factors underlying such errors in an actual clinical setting. Nevertheless, to the best of our knowledge, this survey of claims contains detailed clinical information and is the largest such database in Japan. Fourth, our database only included information from claims in Japan, and it is difficult to generalize the present findings to other countries with different legal systems. Fifth, many cases where settlements were made both in and out of court may not have been included in this database. It is possible that many malpractice cases had been settled privately, without leading to a suit; however, we did not have access to these data. Additionally, outcomes such as minor injuries and mental anguish were not included in this database. Thus, the existing data on malpractice claims in Japan may include a highly select group of medical errors that may not be representative of the general incidence of diagnostic errors. Sixth, the present database can be accessed by purchasing a contract license. For this reason, several important details such as physicians' personal information (e.g., age, sex, postgraduate year, and hospital name) were anonymized, so we could not properly assess these factors. To the best of our knowledge, however, this is the first study investigating diagnostic error-related malpractice claims in Japan, and it utilized the largest internet claims database available. Despite these limitations, the present database remains the most complete source of malpractice data available in the past half century in Japan. Further research on diagnostic errors is needed to better understand the mechanisms underlying diagnostic failures and translate this knowledge into clinical education and patient safety policies.

## Conclusion

In our study, diagnostic errors were a common allegation in malpractice claims and tended to involve allegations of relatively severe patient outcomes compared to other types of malpractice claims. They were also associated with more final court judgments in favor of the claimants, and increased indemnity amounts. To reduce the risk of diagnostic errors, physicians should take care when making diagnoses for potentially serious conditions during general examinations or emergency department visits. A better understanding of malpractice claims might help reduce both patient harm and risk related to physicians' liability.

## Acknowledgments

We express our appreciation to the team members from the Diagnostic Process Improvement Working Group of the Japanese Society of Internal Medicine for sharing their pearls of wisdom with us during this research. We are also grateful to Westlaw Japan K.K. for their kind assistance in extracting the medical malpractice claim data.

## Author Contributions

**Conceptualization:** Takashi Watari, Yasuharu Tokuda, Shohei Mitsuhashi, Hideyuki Kanda.

**Data curation:** Takashi Watari, Shohei Mitsuhashi, Kazuya Otuki, Kaori Kono, Nobuhiro Nagai.

**Formal analysis:** Takashi Watari, Yasuharu Tokuda, Hideyuki Kanda.

**Investigation:** Takashi Watari, Shohei Mitsuhashi, Kazuya Otuki, Kaori Kono, Nobuhiro Nagai.

**Methodology:** Takashi Watari, Yasuharu Tokuda, Hideyuki Kanda.

**Project administration:** Takashi Watari, Yasuharu Tokuda, Shohei Mitsuhashi, Hideyuki Kanda.

**Resources:** Kazumichi Onigata.

**Supervision:** Yasuharu Tokuda, Kazumichi Onigata, Hideyuki Kanda.

**Validation:** Takashi Watari, Yasuharu Tokuda, Shohei Mitsuhashi.

**Writing – original draft:** Takashi Watari.

**Writing – review & editing:** Takashi Watari, Yasuharu Tokuda.

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
