## [Decision Letter · Decision Letter 0]

12 Dec 2019

PONE-D-19-27478

Negative impact and factors of physicians’ diagnostic errors in malpractice claims in Japan

PLOS ONE

Dear Dr. Watari,

Thank you for submitting your manuscript to PLOS ONE. After careful consideration, we feel that it has merit but does not fully meet PLOS ONE’s publication criteria as it currently stands. Therefore, we invite you to submit a revised version of the manuscript that addresses the points raised during the review process.

The paper reports an interesting analysis of the malpractice in Japan. The statistical analysis and results sound but the addition of some details will make it clearer to the readers for example a brief description of the database used in the study providing information useful to evaluate the characteristics of the sample used as highlighted by the reviewers.

We would appreciate receiving your revised manuscript by Jan 26 2020 11:59PM. To enhance the reproducibility of your results, we recommend that if applicable you deposit your laboratory protocols in protocols.io, where a protocol can be assigned its own identifier (DOI) such that it can be cited independently in the future. For instructions see: http://journals.plos.org/plosone/s/submission-guidelines#loc-laboratory-protocols

We look forward to receiving your revised manuscript.

Kind regards,

Lorenza Scotti, PhD

Academic Editor

PLOS ONE

Journal Requirements:

1. Please refer to any post-hoc corrections to correct for multiple comparisons during your statistical analyses. if these were not performed please justify the reasons. Please refer to our statistical reporting guidelines for assistance (https://journals.plos.org/plosone/s/submission-guidelines.#loc-statistical-reporting).

Reviewers' comments:

Reviewer's Responses to Questions

**Comments to the Author**

1. Is the manuscript technically sound, and do the data support the conclusions?

Reviewer #1: Partly

Reviewer #2: Partly

2. Has the statistical analysis been performed appropriately and rigorously? 

Reviewer #1: Yes

Reviewer #2: Yes

3. Have the authors made all data underlying the findings in their manuscript fully available?

Reviewer #1: Yes

Reviewer #2: No

4. Is the manuscript presented in an intelligible fashion and written in standard English?

Reviewer #1: No

Reviewer #2: Yes

5. Review Comments to the Author

Reviewer #1: This is a report that analyzes diagnosis-related cases in a database of Japanese malpractice claims. Over a period of 56 years, they identified 709 cases that involved a diagnostic error, and in this paper they describe the diseases\\conditions involved in these cases, the clinical specialties that were involved, and other descriptive data, in comparison to cases that did not involve diagnostic error.

The methods seem appropriate, the statistical analyses are fine, and results are reasonable and interesting. One of the conclusions, that diagnostic errors are common is not supported and should be removed - the incidence of diagnostic errors was not studied here.

The limitations should emphasize that malpractice claims are a highly select group of errors, and may not be representative of diagnostic errors generally. If the incidence of diagnostic error is anywhere similar to that in the US, there may have been many millions of diagnostic error over the past 56 years, whereas only 709 cases were studied in this analysis.

Minor problems; Please fix these:

Please define ‘acceptance rate’. That isn’t a widely recognized term in the US

Page 5 Lines 67-9 Sentence is unclear. Are cognitive errors the most common etiology in these cases, or are they more common in dx error cases than in other types?

Page 6 Line 78 Do you mean what diagnosis were found in the diagnostic error cases?

Page 12 Line 152 Do you mean that the most common disease category in cases of DERC was neoplasm? They it reads, it sounds like neoplasm was the most common diagnostic error, which is confusing.

Page 15 Line 165 Same issue – the sentence implies these conditions are diagnostic errors; they are not; they are just names of conditions.

Lines 165-9 It isn’t clear why this breakout is important

Page 27 Line 253-5 This sentence isn’t clear. There was no training before 2018? What kind of training?

Page 28 Line 259-60 Do you mean there aren’t enough ER physicians to adequately staff the ER’s? The sentence implies the number of ER’s is greater than the number of MD’s – I doubt it.

Conlusions: The sentence about developing coordinated risk management in exam rooms isn’t clear.

Table 3 I would suggest grouping all the <1% departments together in a “<1%” category instead of listing them separately

Reviewer #2: This paper presents an analysis (primarily descriptive in nature) of Japanese malpractice claims relating to diagnostic errors, with the goal of characterizing the types of injuries and physicians involved and the outcomes of claims. The abstraction and statistical methods appear appropriate. The findings are only of moderate interest, but the paper has a nice discussion section setting them in the context of other literature.

It is not clear how representative the data are of all malpractice claims involving diagnostic errors in Japan, or of all diagnostic errors. It would be helpful to add some background, if possible, regarding how frequently adverse events result in malpractice claims in Japan and what types of selection bias are known to occur. (In the US, for example, only 2-3% of adverse events result in claims and they are disproportionately those involving quite severe injuries). The fact that over half the claims in this sample are death cases and that plaintiffs are, on average, quite young suggests they are not very representative (in the US, at least, medical errors and adverse events disproportionately affect the elderly, and the proportion of malpractice claims involving death is around 20%).

This would provide helpful context for understanding why the number of claims in the sample is so low (1,800 over 56 years, or about 32 per year thoughout the entire country). I do not know anything about the Japanese Westlaw database used, but the authors should provide reassurance that it does indeed contain a comprehensive record of all malpractice claims, as they assert. In the US, all we see in Westlaw are reported malpractice cases—meaning, only the tiny fraction of all claims that result in a trial with a written decision. For that reason, Westlaw databases are considered useless for trying to understand the characteristics of the universe of all malpractice claims, because the small number that go to a bench trial are highly unlikely to be representative of the group. I suspect the Japanese database has the same characteristics, in which case this study should only be published if the authors rewrite it to make clear what they are actually showing: the characteristics of the few cases that make it to a judgment that is reported. They should present contextual information to enable the reader to understand how selective that sample is—specifically, what is known about the percentage of all adverse events that become malpractice claims, and the percentage of all malpractice claims that get to the stage that would permit them to be included in this database.

The study period employed is very long; they look back as far as 1961. Given the vast changes in medicine over the past half century, it seems likely that the characteristics of errors have changed over that period. Therefore, I was surprised not to see any subanalyses of claim characteristics by time period.

The descriptive statistics would also be more interesting if, in addition to the totals, the authors broke them down by accepted and non-accepted claims. Is the median indemnity payment calculated among all claims, or only those that received at least some payment?

6. PLOS authors have the option to publish the peer review history of their article (what does this mean?). If published, this will include your full peer review and any attached files.

Reviewer #1: No

Reviewer #2: No

---

## [Author Response · Author response to Decision Letter 0]

28 Jan 2020

3 January 2020

Dear Dr. Scotti and the PLOS One reviewers:

Thank you for allowing us to revise our paper. The constructive suggestions and feedback provided by the peer reviewers have substantially improved our paper. Attached you will find our detailed responses to their comments. Please note that our revised manuscript contains numerous revisions. Based on the reviewers’ suggestions. 

Specifically, we have added several notable changes based on the reviewers’ suggestion, such as the sub-analyses of changes over study period (1961 to 2010s) and the limitation of selection bias in the extracted dataset. We are submitting the revised version of the manuscript. All of the revisions and changes in the manuscript have been highlighted in yellow. Once again, thank you for your thorough and supportive peer review of our manuscript. 

On the behalf of the authors

Yours sincerely,

Takashi Watari, M.D, MCTM, MS. 

Corresponding author 

Editor 

Editor: The paper reports an interesting analysis of the malpractice in Japan. The statistical analysis and results sound but the addition of some details will make it clearer to the readers for example a brief description of the database used in the study providing information useful to evaluate the characteristics of the sample used as highlighted by the reviewers. 

Authors: We would like to thank editor for your kind comments. Based on the reviewers’ feedback and suggestions, we have revised the manuscript in several places. In particular, regarding the issue of data selection bias, research limitations, and some areas of the discussion points were refined to ensure that conclusions were not overstated and did not mislead the reader. According to data from the Japanese Supreme Court, the number of medical cases conducted in Japan is extremely low compared to rates reported in data from the US, and it is expected that actual adverse events will rarely evolve into lawsuits. This represents a limitation of our study although the data of WestLaw Japan which we used extracted up to one-fifth of the malpractice claims in the 1990s, the rest of the data from the 10-year periods might have selection biases as a reviewer pointed out. However, to data, no comparable study has been conducted in Japan, and we believe that this is the first study that has access to country-based data making it relevant and appropriate for publication in your journal. 

Reviewer 1

Reviewer 1: This is a report that analyzes diagnosis-related cases in a database of Japanese malpractice claims. Over a period of 56 years, they identified 709 cases that involved a diagnostic error, and in this paper they describe the diseases conditions involved in these cases, the clinical specialties that were involved, and other descriptive data, in comparison to cases that did not involve diagnostic error. The methods seem appropriate, the statistical analyses are fine, and results are reasonable and interesting. One of the conclusions, that diagnostic errors are common is not supported and should be removed - the incidence of diagnostic errors was not studied here.

Authors: 

Thank you for your encouraging and supportive comments. We agree and would like to apologize for any imprecision regarding the incidence of diagnostic error, as we believe that it would be more accurate not to report the on the general frequency of diagnostic error in our study. Hence, we have removed the first sentence in the Conclusions section and revised the text to read, “In our study, diagnostic errors in the malpractice claims data from Japan were common and tended to result in poor patient outcomes, an increase in final court judgements for the claims, and increased indemnity amounts.” (P19, L319–321)

Reviewer 1: The limitations should emphasize that malpractice claims are a highly select group of errors, and may not be representative of diagnostic errors generally. If the incidence of diagnostic error is anywhere similar to that in the US, there may have been many millions of diagnostic error over the past 56 years, whereas only 709 cases were studied in this analysis.

Authors: 

We agree with your assessment regarding the selection bias. Thus, we have revised the text as follows: 

P 7, L134–140

In the extracted data, the frequency of the malpractice claims was divided every 10 years and showed the number of claims in each time period: before 1970 (n = 198, 11.0%); in the 1970s (n = 393, 21.8%); in the 1980s (n=366, 20.3%); in the 1990s (n=623, 34.6%); in the 2000s (n = 182, 10.1%); and in the 2010s (n = 30, 1.7%). Although we collected all available malpractice claim cases from the web database, most data represented cases that occurred after the 1970s (n = 1594, 88%). 

P 18–19, L 289–297

In addition, there is no existing data on the frequency with which adverse events lead to malpractice claims in Japan. According to the Japanese Supreme Court report, the total number of adjudicated medical lawsuits in Japan, including those heard in brief and district courts, was not large. There were 305 malpractice claims that received their final judgements in 2000, 324 cases in 2010, and 269 cases in 2016. Furthermore, there are possibly much more than settle cases before suit by patient or their family, unfortunately we could not reach these nationwide data. Thus, the existing malpractice claims in Japan may represent a highly select group of medical errors that may not be representative of the general incidence of diagnostic errors. 

Reviewer 1: Please define ‘acceptance rate’. That isn’t a widely recognized term in the US

Authors: We would like to apologize for using an unclear term, we performed native check again and revised it from “acceptance rate” to “rate of claims with a final court judgment.” These changes are shown in yellow highlights. In addition, we have explained this phrase to refer to the cases in which a judgement that allows a claim seeking a settlement amount (at least partially accepted) has become final and binding. 

Reviewer 1: Page 5 Lines 67-9 Sentence is unclear. Are cognitive errors the most common etiology in these cases, or are they more common in dx error cases than in other types?

Authors: Thank you for calling this to our attention. We have revised this sentence in accordance with your feedback (Page 3, Lines 67), to “…revealed that cognitive errors were the most common cause of these medical claims.” 

Reviewer 1: Page 6 Line 78 Do you mean what diagnosis were found in the diagnostic error cases?

Authors: We apologize any confusion regarding our intended meaning, we collected the data for the first diagnosis, which was referred to as the “initial diagnosis.” This represents when the doctor conducted the first consultation with or examination of the patient. There are many previous reports regarding the final diagnosis and diagnostic error, However, we believe that the initial diagnosis are more critical in diagnostic error research, as it is important to avoid error in actual clinical practice and to record when it occurs. Our review of claim data includes a very detailed analysis of clinical time course in malpractice claims; hence, we believe this represents one of our paper’s notable strengths. We were able to successfully extract the initial diagnosis for almost every malpractice claims before we analyzed the frequency of an initial diagnosis for each diagnostic error. 

Reviewer 1: Page 12 Line 152 Do you mean that the most common disease category in cases of DERC was neoplasm? They it reads, it sounds like neoplasm was the most common diagnostic error, which is confusing. 

Authors: Thank you for pointing this out. We have revised the sentence has been changed in accordance your suggestion (Page 8 from Lines 159–164), to “The two most common initial diagnoses of DERC-involved patients were malignant neoplasms (n = 65, 9.2%) and traumatic injury (n = 64, 8.7%). Further, the five most common malignant diseases were gastric cancer (n = 24; 16%), colorectal cancer (n = 22; 14.7%), breast cancer (n = 16; 10.7%), liver cancer (n = 14; 9.3%), and lung cancer (n = 10; 6.7%). However, these diseases did not occur at a significantly higher frequency than did the non-DERC cases. ”

Reviewer 1: Page 15 Line 165 Same issue – the sentence implies these conditions are diagnostic errors; they are not; they are just names of conditions. 

Authors: Thank you for bringing this to our attention. As you pointed out, they represent actual diagnoses; however, as we mentioned before we used first diagnosis of the malpractice case by reviewing the diagnoses at every clinical time point. Thus, these conditions are the first diagnoses made by doctors that are noted in the claim case information. We have revised part of this sentence based on your feedback (Page 10 from Lines 176), to “Importantly, upper respiratory infections, such as the common cold, acute bronchitis, and pharyngitis, were the most common diagnostic errors when the initial diagnosis was a mild respiratory infection at the time of the first consultation (n = 48/77; 62.3%).”

Reviewer 1: Lines 165-9 It isn’t clear why this breakout is important. 

Authors: Thank you for your constructive question, we regret that the importance of this description was unclear. We categorized some of the common digestive tract diseases, such as gastroenteritis, intestinal obstruction, constipation, as a non-bleeding digestive tract disease. However, gastroenteritis and intestinal obstruction are two very different types of disease, and we believe that the reader might be interested in what the doctor’s first diagnosis in gastrointestinal disease were. Hence, we think this information is important to support the overall findings of our study. 

Reviewer 1: Page 27 Line 253-5 This sentence isn’t clear. There was no training before 2018? What kind of training?

Authors: 

Thank you for your comment, as you pointed out, it could be very difficult for readers who deal with other medical and judicial systems to understand. It is surprisingly that we did not have an official fostering system for general physicians and family medicine, as exists in the training programs in the US. The certified board system in Japan just started in 2018. We have revised this sentence to be more clear based on your feedback (Page 17from Lines 261–263), to “Second, the system of training and supporting generalists, such as general practitioners or family physicians, has only recently begun in Japan, as it started in 2018.”

Reviewer 1: Page 28 Line 259-60 Do you mean there aren’t enough ER physicians to adequately staff the ER’s? The sentence implies the number of ER’s is greater than the number of MD’s – I doubt it.

Authors: 

Thank you for your comment. We apologize for any confusion regarding the emergency medical care situation in Japan. As you noted with some surprise, the total number of emergency room in Japan (at least 4166 in 2005) is much larger than the total number of emergency physicians who are board certificated by the Japanese Association of Acute Care (ER physicians), which were only approximately 4500 in 2016. Thus, most non-emergency physicians who work at medium-sized hospitals are required to serve in the capacity of emergency physicians, regardless of their specialty without training as emergency physician. We believe that this is a critical issue for the Japanese emergency care system and may be one of the causes of diagnostic-error-related malpractice claims. Based on your feedback, we have added further details regarding the number of hospitals and the number of emergency physician in Japan, “The total number of hospitals and beds per national population is the largest in Japan worldwide. Overall, Japan has more than 4000 emergency room; however, the number of emergency physicians who have been certified by the board of emergency care in Japan is quite low (approximately 4500). [30] Thus, an imbalance in the number of hospitals and emergency physicians has occurred. Compared to other high-income countries, non-emergency physicians who are working as surgeon or internist at small- to medium-sized hospital in Japan are also required to work as emergency physicians, regardless of their specialty.” (P 17–18, L 266–273).

Reviewer 1: Conlusions: The sentence about developing coordinated risk management in exam rooms isn’t clear.

Authors: We apologize any lack of clarity. We have removed the sentence, and added a summary sentence, “ To reduce the risk of diagnostic errors, physicians should take care when making diagnoses for potentially serious conditions during a general examination or in emergency department visits.” 

Reviewer 1: Table 3, I would suggest grouping all the <1% departments together in a “<1%” category instead of listing them separately. 

Authors: Thank you for your constructive comment, we have followed your recommendation and have grouped all the <1% departments together in Table 3. 

Reviewer 2

Reviewer 2: This paper presents an analysis (primarily descriptive in nature) of Japanese malpractice claims relating to diagnostic errors, with the goal of characterizing the types of injuries and physicians involved and the outcomes of claims. The abstraction and statistical methods appear appropriate. The findings are only of moderate interest, but the paper has a nice discussion section setting them in the context of other literature. It is not clear how representative the data are of all malpractice claims involving diagnostic errors in Japan, or of all diagnostic errors. It would be helpful to add some background, if possible, regarding how frequently adverse events result in malpractice claims in Japan and what types of selection bias are known to occur. (In the US, for example, only 2-3% of adverse events result in claims and they are disproportionately those involving quite severe injuries). The fact that over half the claims in this sample are death cases and that plaintiffs are, on average, quite young suggests they are not very representative (in the US, at least, medical errors and adverse events disproportionately affect the elderly, and the proportion of malpractice claims involving death is around 20%). This would provide helpful context for understanding why the number of claims in the sample is so low (1,800 over 56 years, or about 32 per year thoughout the entire country). I do not know anything about the Japanese Westlaw database used, but the authors should provide reassurance that it does indeed contain a comprehensive record of all malpractice claims, as they assert. In the US, all we see in Westlaw are reported malpractice cases—meaning, only the tiny fraction of all claims that result in a trial with a written decision. For that reason, Westlaw databases are considered useless for trying to understand the characteristics of the universe of all malpractice claims, because the small number that go to a bench trial are highly unlikely to be representative of the group. I suspect the Japanese database has the same characteristics, in which case this study should only be published if the authors rewrite it to make clear what they are actually showing: the characteristics of the few cases that make it to a judgment that is reported. They should present contextual information to enable the reader to understand how selective that sample is—specifically, what is known about the percentage of all adverse events that become malpractice claims, and the percentage of all malpractice claims that get to the stage that would permit them to be included in this database.

Authors: We would like to thank the reviewer for these kind comments. We apologize for being unclear how representative the data are of all malpractice claims involving diagnostic errors in Japan, or of all diagnostic errors. Unfortunately, despite our careful literature review, there is no existing nation-level data regarding the frequency with which adverse events lead to malpractice claims in Japan. Although malpractice claims have shown a slight increase slightly recently, this rate seems to be far less than that of the United States. In fact, the Japanese Supreme Court has reported that the number of the total medical lawsuits in Japan (including brief and district courts) included only 305 claims with final judgements in 2000, 324 in 2010, and 269 cases in 2016. There are possibly many more cases that are settled before a legal proceedings are begun by patients or their family; however, we do not have access to these kind of the data. The reason the median age of the patients are relatively young might be due to the high number of stillbirths of newborns and sudden deaths of children. Especially in Japan, deaths in childhood are more likely to result in malpractice litigation. While the absolute number of elderly patients living in Japan is certainly large, the rate of malpractice claims for this population that leads to litigation is small

We used an Internet database provided by WestLaw Japan. We believe this datasource does have a selection bias as the reviewers have pointed out. As we mentioned earlier, the number of cases of medical litigation in Japan is not as large as in the United States. The reason for this is unclear, but it could be due to the cultural background in Japan which leads to patients and family unlikely to appeal to physicians due to the traditional Japanese culture of reconciliation, as well as a legal system that doesn't provide a large financial benefit for patients to sue a healthcare professionals. Since the total number of adjudicated medical lawsuits during the decade of the 1990s was about 3,000 cases in nationwide, we had 623, representing one-fifth of the total number of cases from the 1990s. Since the 2000s, there has yet been full transcription from paper-based files; hence, the amount of claim data were small. However, as Japanese first nationwide report of malpractice claims about DERC, the data source represents the only available data in Japan that allows researchers to extract precise and detail information for detailed case.

We have added a limitation to clarify the context of our data and identify the possibility of selection bias of this study. 

“In addition, there is no existing data on the frequency with which adverse events lead to malpractice claims in Japan. According to the Japanese Supreme Court report, the total number of adjudicated medical lawsuits in Japan, including those heard in brief and district courts, was not large. There were 305 final judged malpractice claims in 2000, 324 cases in 2010, and 269 cases in 2016. Furthermore, there are possibly much more than settle cases before suit by patient or their family, unfortunately we could not reach these nationwide data. Thus, the existing malpractice claims in Japan may represent a highly select group of medical errors that may not be representative of the general incidence of diagnostic errors.” (P 18–19, L289–297)

Reviewer 2: The study period employed is very long; they look back as far as 1961. Given the vast changes in medicine over the past half century, it seems likely that the characteristics of errors have changed over that period. Therefore, I was surprised not to see any subanalyses of claim characteristics by time period.

Authors: Thank you for raising this important point. We entirely agree that these sub-analyses have merit and apologize for being unclear about this. We did not report the characteristics of the diagnostic error-related malpractice claims over time. We initially examined which period had the most extracted data with the histogram, which showed that malpractice claims were the most frequent in the 1990s. This might be owing to the spread of the Internet in Japan that promoted the sharing of the data online since 1990.

Then, we divided the data into 10 year periods (1944–1950, 1950s, 1960s, 1970s, 1980s, 1990s, 2000s, 2010–2014), and compared the DERC cases and non-DERC cases over each 10-year period. As a result, each DERC proportion in the 10-year periods showed between 1944–1950, 33.3% p = 0.669; the 1950s, 25.0% p = 0.117;the 1960s, 26.79% p < 0.001; the 1970s, 37.4% p = 0.373; the 1980s, 22.1% p = 0.119; the 1990s, 34.1% p = 0.1001; the 2000s, 52.0% p < 0.001; 2010–2014, 40% p = 0.941. We also performed a multivariate logistic analysis to compare the DERC and non-DERC groups over each 10-year period. This analysis did not show a significant proportion of DERC among each 10 years. However, as you suggested about these sub-analyses by time period, we definitely believe reviewer’s comment is very appropriate and important for reader to understand. We added the information as follows, “In the extracted data, the frequency of the malpractice claims was divided every 10 years and showed the number of claims in each time period: before 1970 (n = 198, 11.0%); in the 1970s (n = 393, 21.8%); in the 1980s (n=366, 20.3%); in the 1990s (n=623, 34.6%); in the 2000s (n = 182, 10.1%); and in the 2010s (n = 30, 1.7%). Although we collected all available malpractice claim cases from the web database, most data represented cases that occurred after the 1970s (n = 1594, 88%). Each DERC proportion for each 10-year period showed that only the period before 1970 and the 2000s were significantly different (26.79% p-value<0.001, 2000s; 52.0% p-value<0.001). We also performed a multivariate logistic analysis to compare the DERC and non-DERC groups of DERC over each 10-year period and found no significant proportion of DERC among each 10-year period.” (P 6–7, L 134–144)

Reviewer 2: The descriptive statistics would also be more interesting if, in addition to the totals, the authors broke them down by accepted and non-accepted claims. 

Authors: Thank you for your feedback, as you have pointed out, we believe that the comparison study of accepted and non-accepted malpractice claims would be very interesting. However, our primary goal for this manuscript was to clearly describe diagnostic error in malpractice claims and the contributing factors to diagnostic error in existing malpractice claims in Japan. After this, we would like to procced with a comparison study in near future. We really appreciate your constructive guidance. 

Reviewer 2: Is the median indemnity payment calculated among all claims, or only those that received at least some payment?

Authors: Thank you for bringing up this important point, and agree that the definition of this value should be clarified. We used the two words to mean the same thing (Median indemnity payment = the adjusted median in the final judgement). The median indemnity payment was among only claims received at least some payment. Hence, we changed the median indemnity payment to “the adjusted median for the final judgement amount was $183,6367 (n = 941, IQR = $41.462–$440,909), with a 52.6% of claims having a final judgement.” in P7 Line 147-149 & Table 1. Thank you.

---

## [Decision Letter · Decision Letter 1]

3 Mar 2020

PONE-D-19-27478R1

Negative impact and factors of physicians’ diagnostic errors in malpractice claims in Japan

PLOS ONE

Dear Dr. Watari,

Thank you for submitting your manuscript to PLOS ONE. After careful consideration, we feel that it has merit but does not fully meet PLOS ONE’s publication criteria as it currently stands. Therefore, we invite you to submit a revised version of the manuscript that addresses the points raised during the review process.

Although the manuscript is improved after addressing reviewer's suggestions, a few minor revisions are still needed.

We would appreciate receiving your revised manuscript by Apr 17 2020 11:59PM. To enhance the reproducibility of your results, we recommend that if applicable you deposit your laboratory protocols in protocols.io, where a protocol can be assigned its own identifier (DOI) such that it can be cited independently in the future. For instructions see: http://journals.plos.org/plosone/s/submission-guidelines#loc-laboratory-protocols

We look forward to receiving your revised manuscript.

Kind regards,

Lorenza Scotti, PhD

Academic Editor

PLOS ONE

Reviewers' comments:

Reviewer's Responses to Questions

**Comments to the Author**

1. If the authors have adequately addressed your comments raised in a previous round of review and you feel that this manuscript is now acceptable for publication, you may indicate that here to bypass the “Comments to the Author” section, enter your conflict of interest statement in the “Confidential to Editor” section, and submit your "Accept" recommendation.

Reviewer #1: (No Response)

Reviewer #2: (No Response)

2. Is the manuscript technically sound, and do the data support the conclusions?

Reviewer #1: Yes

Reviewer #2: Yes

3. Has the statistical analysis been performed appropriately and rigorously? 

Reviewer #1: Yes

Reviewer #2: Yes

4. Have the authors made all data underlying the findings in their manuscript fully available?

Reviewer #1: Yes

Reviewer #2: No

5. Is the manuscript presented in an intelligible fashion and written in standard English?

Reviewer #1: Yes

Reviewer #2: Yes

6. Review Comments to the Author

Reviewer #1: Thank you for addressing my concerns in the revised version - the revised version is much improved as a result.

Reviewer #2: The authors have been reasonably responsive to the reviewer comments. They appear to have missed the point of the comment about separating claims that did and did not result in a payment, however. Because the authors’ aim is to say something about the nature of diagnostic errors using malpractice claims as a data source, their study question is more directly answered by examining claims that are meritorious (i.e. actually do involve an error) than by analyzing all claims, many of which (if data from several other countries are any indication) are nonmeritorious—i.e., do not actually relate to injuries due to an error. In the US, for example, estimates of the proportion of claims that are nonmeritorious range from a third to over half of all claims. That’s why it is especially useful, given the severe limitations of this data source, to isolate claims with the greatest signal about actual errors. Payment is an imperfect proxy measure of that; claims get paid sometimes even when they don’t involve errors. But it’s better than nothing.

Some of the edits made to the manuscript read a bit rough; they could benefit from editing by a native English speaker and from the following revisions to the limitations section:

- The new limitation added on p.18 (“there is no existing data on the frequency with which adverse events lead to malpractice claims in Japan”) should be called out as separate from the first limitation, as it is a wholly different point. This limitation should be labeled “Second,” and the limitations that followed should be renumbered accordingly.

- The sentence beginning “Furthermore, there are possibly much more cases…” on p.19 does not belong where it is inserted. It is again a separate point, and it is the same point the authors already make farther down the page (“Third, many settlements…”).

Finally, the revisions do a good job overall of acknowledging that malpractice claims aren’t a very good source of information about all diagnostic errors, but they could be improved by revising a few remaining instances where the authors talk about errors rather than claims:

- P.16: “about 40% of the claims were considered diagnostic errors” should instead read “about 40% of the claims involved allegations of a diagnostic error”

- P.16: “The initial diagnoses most likely to lead to diagnostic errors were” should read “The diagnoses most commonly associated with allegations of diagnostic errors in malpractice claims were”

- P.20: “diagnostic errors in the malpractice claims data from Japan were common” should instead read “diagnostic errors were a common allegation in malpractice claims”

- P.20: “and tended to result in poor patient outcomes” implies causality. It should instead read “and tended to involve allegations of relatively severe patient outcomes compared to other types of malpractice claims.”

7. PLOS authors have the option to publish the peer review history of their article (what does this mean?). If published, this will include your full peer review and any attached files.

Reviewer #1: No

Reviewer #2: No

---

## [Author Response · Author response to Decision Letter 1]

2 Apr 2020

28 March 2020

Dear Dr. Scotti and the PLOS One reviewers: 

Thank you for allowing us to revise the manuscript. The constructive suggestions and feedback provided by the reviewers have substantially improved our paper. Our revised manuscript contains modifications based on the reviewers’ suggestions. 

Once again, thank you for your thorough and supportive peer review. 

On behalf of the authors, yours sincerely,

Takashi Watari, MD, MCTM, MS 

Corresponding author 

Reviewer #2: 

“The authors have been reasonably responsive to the reviewer comments. They appear to have missed the point of the comment about separating claims that did and did not result in a payment, however. Because the authors’ aim is to say something about the nature of diagnostic errors using malpractice claims as a data source, their study question is more directly answered by examining claims that are meritorious (i.e. actually do involve an error) than by analyzing all claims, many of which (if data from several other countries are any indication) are nonmeritorious—i.e., do not actually relate to injuries due to an error. In the US, for example, estimates of the proportion of claims that are nonmeritorious range from a third to over half of all claims. That’s why it is especially useful, given the severe limitations of this data source, to isolate claims with the greatest signal about actual errors. Payment is an imperfect proxy measure of that; claims get paid sometimes even when they don’t involve errors. But it’s better than nothing.”

Response: 

We understand your comment for this study and wholly agree with your idea that it would be useful information for clinicians. Thus, we added the related information in response to the reviewer’s comment,

P7 L153

 “In addition, we also analyzed the above 941 case (52.6%) of claims having a final judgement. Death was the most common claims outcome (n = 473/941; 50.27%), followed by sequelae (41.8%) and full recovery (6.3%) The median patient age was 32 years (interquartile range [IQR] = 10¬.5–¬¬53), and 54.2% were men. The median claim duration was 7 years (M = 7.64 years, IQR = 5–9 years, maximum = 25 years). A total of 447 (47.5%, 95% confidence interval [CI]: 44.3%–50.7%) DERC cases were observed.”

However, our aim of this study was to better characterize the negative impact of diagnostic error reported in malpractice claims; hence, after this study, we will proceed with a comparison study in near future. We very much appreciate your constructive guidance. 

Reviewer #2: 

“Some of the edits made to the manuscript read a bit rough; they could benefit from editing by a native English speaker and from the following revisions to the limitations section:”

Response:

We thank the reviewer for this pertinent comment. Accordingly, we have changed the following text per the advice of a native-English-speaking editor: 

P6 L134 (only minor changes)

“In the extracted data, malpractice claim frequency was measured using 10-year periods, and the number of claims in each time period was determined: before 1970 (n = 198; 11.0% of total malpractice claims); during the 1970s (n = 393; 21.8%); during the 1980s (n=366; 20.3%); during the 1990s (n = 623; 34.6%); during the 2000s (n = 182; 10.1%); and during the 2010s (n = 30; 1.7%). Although we collected all available malpractice claim cases from the web database, most data represented cases that occurred after the 1970s (n = 1,594; 88%). The DERC percentage for each 10-year period showed only the period before 1970, and the 2000s were significantly different (before 1970: 26.79%, p < 0.001; 2000s: 52.0%, p < 0.001). We also performed a multivariate logistic analysis to compare the DERC and non-DERC groups over each 10-year period and found no significant proportion of DERC among each 10-year period.”

“- The new limitation added on p.18 (“there is no existing data on the frequency with which adverse events lead to malpractice claims in Japan”) should be called out as separate from the first limitation, as it is a wholly different point. This limitation should be labeled “Second,” and the limitations that followed should be renumbered accordingly.”

 and

- The sentence beginning “Furthermore, there are possibly much more cases…” on p.19 does not belong where it is inserted. It is again a separate point, and it is the same point the authors already make farther down the page (“Third, many settlements…”).

 and

- The sentence beginning “Furthermore, there are possibly much more cases…” on p.19 does not belong where it is inserted. It is again a separate point, and it is the same point the authors already make farther down the page (“Third, many settlements…”).

Thank you for raising this important point. We completely agree with the reviewers’ comments; therefore, we have revised the text following these suggestions. 

P18 L294

“Therefore, further training in general and emergency medicine as well as improvement of work conditions are required to minimize diagnostic errors in these settings in Japan.

Several limitations should be noted. First, although we used the largest claims database in Japan (similar to previous studies [12,29,32,33]), the data were not nationally representative of all malpractice claims. Second, there are no existing data on the frequency with which adverse events lead to malpractice claims in Japan. According to the Japanese Supreme Court report, the total number of adjudicated medical lawsuits in Japan, including those heard in brief and district courts, was not large. There were 305 malpractice claims that received their final judgments in 2000, 324 cases in 2010, and 269 cases in 2016. Third, the data from malpractice claims are not direct medical records and are thus not ideal sources for investigating error frequency and causal factors underlying such errors in an actual clinical setting. Nevertheless, to the best of our knowledge, this survey of claims contains detailed clinical information and is the largest such database in Japan. Fourth, our database only included information from claims in Japan, and it is difficult to generalize the present findings to other countries with different legal systems. Fifth, many cases where settlements were made both in and out of court may not have been included in this database.” 

-“Finally, the revisions do a good job overall of acknowledging that malpractice claims aren’t a very good source of information about all diagnostic errors, but they could be improved by revising a few remaining instances where the authors talk about errors rather than claims:”

We would like to thank the reviewer for these kind comments. We apologize for being unclear about how representative the data are of all malpractice claims involving diagnostic errors in Japan. However, thanks to reviewer’s constructive suggestions, we have made these necessary changes to the manuscript. 

- P.16: “about 40% of the claims were considered diagnostic errors” should instead read “about 40% of the claims involved allegations of a diagnostic error”

P16 L242 

Our results based on 1,802 malpractice claims over a 56-year period in Japan showed that about 40% of the claims involved allegations of a diagnostic error

- P.16: “The initial diagnoses most likely to lead to diagnostic errors were” should read “The diagnoses most commonly associated with allegations of diagnostic errors in malpractice claims were”

P16 L245 

The initial diagnoses most commonly associated with allegations of diagnostic errors in malpractice claims were upper respiratory tract infection (mainly common cold), non-bleeding digestive tract disease (mainly gastroenteritis), and “no abnormality.”

- P.20: “diagnostic errors in the malpractice claims data from Japan were common” should instead read “diagnostic errors were a common allegation in malpractice claims”

- P.20: “and tended to result in poor patient outcomes” implies causality. It should instead read “and tended to involve allegations of relatively severe patient outcomes compared to other types of malpractice claims.”

P20 L329

In our study, diagnostic errors were a common allegation in malpractice claims and tended to involve allegations of relatively severe patient outcomes compared to other types of malpractice claims.

Thank you again for your comments regarding our paper. I hope you find the revised manuscript suitable for publication.

---

## [Editor Report · Decision Letter 2]

14 Apr 2020

PONE-D-19-27478R2

Negative impact and factors of physicians’ diagnostic errors in malpractice claims in Japan

PLOS ONE

Dear Dr. Watari,

Thank you for submitting your manuscript to PLOS ONE. After careful consideration, we feel that it has merit but does not fully meet PLOS ONE’s publication criteria as it currently stands. Therefore, we invite you to submit a revised version of the manuscript that addresses the points raised during the review process.

Although the manuscript has been improved after the first revision, a few points raised by Reviewer 2 need to be addressed.

We would appreciate receiving your revised manuscript by May 29 2020 11:59PM. To enhance the reproducibility of your results, we recommend that if applicable you deposit your laboratory protocols in protocols.io, where a protocol can be assigned its own identifier (DOI) such that it can be cited independently in the future. For instructions see: http://journals.plos.org/plosone/s/submission-guidelines#loc-laboratory-protocols

We look forward to receiving your revised manuscript.

Kind regards,

Lorenza Scotti, PhD

Academic Editor

PLOS ONE

---

## [Author Response · Author response to Decision Letter 2]

28 May 2020

May 28 2020

Dear Dr. Scotti and the PLOS One reviewers:

Thank you for allowing us to revise the manuscript. The constructive suggestions and feedback provided by the reviewers have substantially improved our paper. Our revised manuscript contains modifications based on the reviewers’ suggestions.

Once again, thank you for your thorough and supportive peer review.

On behalf of the authors, yours sincerely,

Takashi Watari, MD, MCTM, MS

Corresponding author

Reviewer #2: 

“The authors have been reasonably responsive to the reviewer comments. They appear to have missed the point of the comment about separating claims that did and did not result in a payment, however. Because the authors’ aim is to say something about the nature of diagnostic errors using malpractice claims as a data source, their study question is more directly answered by examining claims that are meritorious (i.e. actually do involve an error) than by analyzing all claims, many of which (if data from several other countries are any indication) are nonmeritorious—i.e., do not actually relate to injuries due to an error. In the US, for example, estimates of the proportion of claims that are nonmeritorious range from a third to over half of all claims. That’s why it is especially useful, given the severe limitations of this data source, to isolate claims with the greatest signal about actual errors. Payment is an imperfect proxy measure of that; claims get paid sometimes even when they don’t involve errors. But it’s better than nothing.”

Response: 

I wholeheartedly agree with your view that the comparison of meritorious trials alone would provide the most useful answer for our study question, and we have to apologize for the misunderstanding caused by the contextual differences between Japanese and US legal constructs. For example, “Claims with final judgment resulting in payment” (Table 1 in the revised manuscript) refers to cases where judgment was in favor of the patient, resulting in a payment order for the medical practitioner. We have revised the manuscript to clarify our meaning.

I do believe there is a great difference between the way medical litigation is practiced in the US compared to Japan. There are only about 800 medical litigation cases in Japan each year, most of which are meritorious. While we realize that cultural practices cannot be quantified as evidence, it is important to note that Japanese people are generally reluctant to dispute healthcare-related decisions. This means that trials usually only result from cases where there is an obvious problem in the care provided by medical staff. On the other hand, the number of medical litigation cases in the US is about 17,000 a year – 21 times more than the number of cases in Japan. Furthermore, the number of medical litigation cases per person is 7.7 times more in the US, compared to Japan. Considering these contextual differences, if this study was conducted in the US and included all medical litigation, the data would include an overwhelmingly high number of unmeritorious cases, and be strongly biased. All the cases in our dataset had been concluded, and clarified the validity of claims in detail. We do not consider those cases not resulting in payment to be unmeritorious, and we conclude that there are currently few unmeritorious medical cases under our judicial system.

Second, under the supervision of the third author, a lawyer, we narrowed our dataset down to 1,802 meritorious medical cases that deserve analysis (Figure 1). This work took an enormous amount of time and ruled out any non-meritorious trials to eliminate bias as much as possible. Notably, of the approximately 1,900 medical litigation cases, only 34 were classified as unjustified and unjust by the third author. As noted previously, unfair trials in medical litigation are rare in Japan.

Third, the frequency of doctors’ diagnostic errors in actual medical practice is high. In other words, if we only analyze the case when the doctor loses and payment is required, the actual diagnostic error leading to the winning lawsuit would be ignored, even though the doctor had made a diagnostic error. Our concern was that we would exclude winning cases where diagnostic errors existed, as medical errors are difficult to prove in Japanese medical law, resulting in low-cost medical professional liability insurance; thus, even when clear diagnostic errors exist, it would not necessarily lead to medical litigation.

I believe your opinion is correct in the context of US medical law, and our next goal is to conduct a second analysis based on your advice. I really appreciate your constructive peer review.

Changes

P7 L153

“In addition, we specifically analyzed the 941 claims where final judgment resulted in payment. Among these, death was the most common claim outcome (n = 473/941; 50.27%), followed by sequelae (41.8%) and full recovery (6.3%). The median patient age was 32 years (interquartile range [IQR] = 10¬.5–¬¬53), and 54.2% were men. The median claim duration was 7 years (M = 7.64 years, IQR = 5–9 years, maximum = 25 years). Of these, 447 claims (47.5%, 95% confidence interval [CI]: 44.3%–50.7%) were DERC cases. ”

Reviewer #2: 

“Some of the edits made to the manuscript read a bit rough; they could benefit from editing by a native English speaker and from the following revisions to the limitations section:”

Response:

We thank the reviewer for this pertinent comment. Accordingly, we have revised the following text per the advice of a native-English-speaking editor:

(only minor changes)

P6 L133 

“In the extracted data, malpractice claim frequency was measured using 10-year periods, and the number of claims in each period was determined: before 1970 (n = 198; 11.0% of total malpractice claims), during the 1970s (n = 393; 21.8%), during the 1980s (n=366; 20.3%), during the 1990s (n = 623; 34.6%), during the 2000s (n = 182; 10.1%), and during the 2010s (n = 30; 1.7%). Although we collected all available malpractice claim cases from the database, most data represented cases that occurred after the 1970s (n = 1,594; 88%). The DERC percentage for each 10-year period was significantly different only for the period before 1970 and during the 2000s (before 1970: 26.79%, p < 0.001; 2000s: 52.0%, p < 0.001). We also performed a multivariate logistic analysis to compare the DERC and non-DERC groups over each 10-year period and found no significant proportion of DERC for any period.”

Reviewer #2: 

“- The new limitation added on p.18 (“there is no existing data on the frequency with which adverse events lead to malpractice claims in Japan”) should be called out as separate from the first limitation, as it is a wholly different point. This limitation should be labeled “Second,” and the limitations that followed should be renumbered accordingly.”

 and

- The sentence beginning “Furthermore, there are possibly much more cases…” on p.19 does not belong where it is inserted. It is again a separate point, and it is the same point the authors already make farther down the page (“Third, many settlements…”).

Thank you for raising these important points. We completely agree with the reviewers’ comments; therefore, we have revised the text following these suggestions. 

Changes

P18 L294

“Therefore, further training in general and emergency medicine, along with improvement of work conditions, are required to minimize diagnostic errors in these settings in Japan. However, several limitations of this study should be noted. First, although we used the largest claims database in Japan (similar to previous studies [12,29,32,33]), the data were not nationally representative of all malpractice claims. Second, there are no existing data on the frequency with which adverse events lead to malpractice claims in Japan. According to the Japanese Supreme Court report, the total number of adjudicated medical lawsuits in Japan, including those heard in brief and district courts, was not large [13]: there were 305 malpractice claims that received their final judgments in 2000, 324 in 2010, and 269 in 2016. Third, the data from malpractice claims are not direct medical records and are thus not ideal sources for investigating error frequency and causal factors underlying such errors in an actual clinical setting. Nevertheless, to the best of our knowledge, this survey of claims contains detailed clinical information and is the largest such database in Japan. Fourth, our database only included information from claims in Japan, and it is difficult to generalize the present findings to other countries with different legal systems. Fifth, many cases where settlements were made both in and out of court may not have been included in this database.

Reviewer #2: 

-“Finally, the revisions do a good job overall of acknowledging that malpractice claims aren’t a very good source of information about all diagnostic errors, but they could be improved by revising a few remaining instances where the authors talk about errors rather than claims:”

We would like to thank the reviewer for these kind comments. We apologize for being unclear about how representative the data are of all malpractice claims involving diagnostic errors in Japan. However, thanks to the reviewer’s constructive suggestions, we have made necessary changes to the manuscript.

Changes

P16 L244 

Our results, based on 1,802 malpractice claims over a 56-year period in Japan, showed that about 40% of the claims involved allegations of a diagnostic error, that

Reviewer #2: 

- P.16: “The initial diagnoses most likely to lead to diagnostic errors were” should read “The diagnoses most commonly associated with allegations of diagnostic errors in malpractice claims were”

Changes

P16 L24 

The initial diagnoses most commonly associated with allegations of diagnostic errors in malpractice claims were upper respiratory tract infection (mainly the common cold), non-bleeding digestive tract disease (mainly gastroenteritis), and “no abnormality.”

Reviewer #2: 

- P.20: “diagnostic errors in the malpractice claims data from Japan were common” should instead read “diagnostic errors were a common allegation in malpractice claims”

- P.20: “and tended to result in poor patient outcomes” implies causality. It should instead read “and tended to involve allegations of relatively severe patient outcomes compared to other types of malpractice claims.”

Changes

P20 L328

In our study, diagnostic errors were a common allegation in malpractice claims and tended to involve allegations of relatively severe patient outcomes compared to other types of malpractice claims.

Thank you again for your feedback on our paper. I hope you find the revised manuscript suitable for publication.

---

## [Editor Report · Decision Letter 3]

22 Jul 2020

Factors and impact of physicians’ diagnostic errors in malpractice claims in Japan

PONE-D-19-27478R3

Dear Dr. Watari,

We’re pleased to inform you that your manuscript has been judged scientifically suitable for publication and will be formally accepted for publication once it meets all outstanding technical requirements.

Kind regards,

Cesario Bianchi

Academic Editor

PLOS ONE

Additional Editor Comments (optional):

Dear Dr Watari,

Thank you for carefully revising our manuscript according to reviewer #2 critics.

Your manuscript is now acceptable for publication in our journal.
---

## [Editor Report · Acceptance letter]

23 Jul 2020

PONE-D-19-27478R3 

Factors and impact of physicians’ diagnostic errors in malpractice claims in Japan 

Dear Dr. Watari:

I'm pleased to inform you that your manuscript has been deemed suitable for publication in PLOS ONE. Congratulations! Your manuscript is now with our production department. 

Kind regards, 

on behalf of

Dr. Cesario Bianchi 

Academic Editor

PLOS ONE